# Electrospinning of Cellulose Nanocrystal-Reinforced Polyurethane Fibrous Mats

**DOI:** 10.3390/polym12051021

**Published:** 2020-05-01

**Authors:** Alexandre Redondo, Daseul Jang, LaShanda T. J. Korley, Ilja Gunkel, Ullrich Steiner

**Affiliations:** 1Adolphe Merkle Institute, University of Fribourg, Chemin des Verdiers 4, 1700 Fribourg, Switzerland; alexandre.redondo@unifr.ch; 2Department of Materials Science and Engineering, University of Delaware, Newark, DE 19716, USA; dsjang@udel.edu (D.J.); lkorley@udel.edu (L.T.J.K.); 3Department of Chemical and Biomolecular Engineering, University of Delaware, Newark, DE 19716, USA

**Keywords:** polyurethane/CNC nanocomposites, electrospinning, fiber alignment, mechanical properties, reinforcement

## Abstract

We report the electrospinning of mechanically-tunable, cellulose nanocrystal (CNC)-reinforced polyurethanes (PUs). Using high-aspect ratio CNCs from tunicates, the stiffness and strength of electrospun PU/CNC mats are shown to generally increase. Furthermore, by tuning the electrospinning conditions, fibrous PU/CNC mats were created with either aligned or non-aligned fibers, as confirmed by scanning electron microscopy. PU/CNC mats having fibers aligned in the strain direction were stiffer and stronger compared to mats containing non-aligned fibers. Interestingly, fiber alignment was accompanied by an anisotropic orientation of the CNCs, as confirmed by wide-angle X-ray scattering, implying their alignment additionally benefits both stiffness and strength of fibrous PU/CNC nanocomposite mats. These findings suggest that CNC alignment could serve as an additional reinforcement mechanism in the design of stronger fibrous nanocomposite mats.

## 1. Introduction

Electrospinning as a means to produce polymer fibers from solution was first introduced in the early 20th century and was patented by Formhas in 1934 [1,2]. Long forgotten, it was rediscovered in the early 1990s by academia and industry during the advent of nanofiber technology [3]. Nanofibers have the potential to significantly improve existing technologies where samples with a high specific area, high porosity, or low pore size are beneficial. Electrospinning enables the production of fibers with different morphologies (fibers, hollow fibers, beaded fibers, etc.), porosity, and fiber diameters from a few nanometers to several micrometers. Electrospun mats are useful for a wide range of potential applications including filtration, batteries, fuel cells, tissue scaffolds, nano-catalysis, wound dressing, and membrane technology [3,4,5,6]. However, due to their non-woven nature, electrospun mats are mechanically weak, limiting their use in certain applications, such as membrane filtration [1,7,8,9]. One strategy to fabricate stronger electrospun mats is the addition of cellulose-based fillers to the polymeric matrix before electrospinning, which not only provides effective reinforcement but is also ecological, environmentally friendly, and biocompatible. Cellulose, the most abundant biopolymer on Earth, can be extracted from plants, bacteria, and tunicates. It can be processed into various reinforcing agents such as microcrystalline cellulose (MCC) [10,11,12], cellulose nanofibers (CNFs) [12,13,14] and cellulose nanocrystals (CNCs) that differ in their aspect ratio and crystallinity [15,16,17].

CNCs have recently attracted much attention as polymer reinforcement agents, leading to CNC-based composites with extraordinary mechanical properties. CNCs have a high specific surface area, low densities, and high stiffnesses, ranging 110–220 GPa in the longitudinal and 10–50 GPa in the transverse directions [18]. CNCs have typical diameters of 5–50 nm and lengths ranging from a few hundred nanometers for cotton CNCs to three micrometers for CNCs extracted from tunicates [15,19,20]. Among all cellulose sources, tunicate CNCs (t-CNCs) have been shown to possess the highest aspect ratio, leading to more effective mechanical reinforcement of polymer matrices, compared to CNCs obtained from other sources [20,21,22].

Among the various available polymer matrices, polyurethanes (PUs) have received particular attention. This is reflected in numerous studies involving PU/CNC nanocomposites, reporting strong mechanical reinforcement [23,24,25,26,27,28,29]. Most PUs are microphase-separated thermoplastic elastomers consisting of a rubbery phase formed by a non-crystallizing telechelic polymer with a low glass transition temperature $T_{g}$ and hard segments composed of diisocyanate and diols that serve as physical cross-links. PUs typically offer high strength and elasticity, as well as molecular tunability [30,31]. PU/CNC nanocomposites can be processed by melt-spinning [28,32], melt-mixing followed by compression molding [21,22,23,24,25,26,27], and solvent casting [23,24,25,33,34]. PU/CNC nanocomposites electrospun into fibrous mats were only demonstrated very recently [35,36]. Electrospun CNC-reinforced fibrous PU mats are, however, promising for a wide range of applications, for example in tissue engineering and membrane filtration [9]. Electrospun CNC-based nanocomposites have also been demonstrated employing polystyrene [37,38], poly(lactic acid) [39], polycaprolactone [40], and poly(ethylene oxide) [41,42,43] matrices. Most studies focused mainly on the effect of various processing parameters on the diameter and morphology of the fibers within electrospun mats, while only very few were concerned with enhancing the mechanical properties of the mats through fiber alignment or CNC orientation. In particular, the contribution of CNC alignment to the reinforcement of (aligned) fibrous mats has only been scarcely studied. Progress in this direction helps to identify design strategies for stronger fibrous nanocomposite mats and increase their utility.

Here, we demonstrate the electrospinning of nanocomposite fibrous mats comprised of a PU matrix reinforced with t-CNCs. Using different collector configurations, electrospun mats were produced with either aligned or non-aligned fibers, which were mechanically evaluated by tensile testing. We show that the mechanical properties of electrospun fibrous mats improve upon CNC addition, which is further enhanced by fiber alignment. We describe the influence of CNC addition on the fiber morphology and diameter. The CNC orientation within the mats was determined by wide-angle X-ray scattering (WAXS).

## 2. Materials and Methods

### 2.1. Materials

Texin 985, a thermoplastic polyurethane (PU) based on poly(tetramethylene glycol), butanediol, and 4,4${}^{\prime }$-methylenebis(phenyl isocyanate) with a Shore A hardness of about 85, was obtained from Covestro [44]. Dimethylformamide (DMF) and phosphoric acid 85% were obtained from Sigma-Aldrich Corp. and were used as received without further purification.

### 2.2. Isolation of Cellulose Nanocrystals (P-tCNCs) from Tunicates by Phosphoric Acid Hydrolysis

Tunicates collected from the French Atlantic coast were purified using Soxhlet extraction and subsequent bleaching following procedures described elsewhere [45]. The extracted material was dried and ground into a powder using a HR2195/00 blender from Philips (PHILIPS, Amsterdam, Netherlands) at maximum speed (900 W output power). The hydrolysis conditions were adapted from the procedure described by Camarero Espinosa et al. for cotton CNCs [15]. A 75 wt% phosphoric acid solution was prepared by adding 412 mL of 85 wt% phosphoric acid to 88 mL of deionized water under vigorous stirring. The acid solution was heated to 100 °C, before 2 g of the ground, dried, and bleached tunicate mantles were added, followed by stirring for 4 h at 100 °C. The mixture was then cooled to 25 °C by the addition of 500 g of ice and centrifuged at 14,500 RPM for 30 min. The supernatant was decanted, replaced with deionized water, and the centrifugation step was repeated at least two times until the supernatant was transparent and colorless. The remaining suspension was then dialyzed against deionized water for five days, exchanging the water every day. The final P-tCNC suspension, having a pH of about 5.5–6, was redispersed by sonication for 15 min using a horn ultrasonicator (Branson Digital Sonifier S-250D, Branson Ultrasonics, Danbury, CT, USA, 50–60 Hz/200 W) at 15% amplitude. The dispersed suspension was frozen overnight in a conventional freezer and lyophilized (Telstar LyoQuest Laboratory Freeze Dryer, Terassa, Spain) for three days. The procedure yielded P-tCNCs as a white cotton-like material with a yield of 55–65%.

### 2.3. Electrospinning of Nanocomposite Fibers

First, 12, 60, and 120 mg of P-tCNCs were dispersed separately into 20 mL DMF using an ultrasonic bath for 2 h. To each P-tCNC/DMF solution, 1.2 g of PU (Texin^®^ 985) were then added to achieve nanocomposites with concentrations of 1, 5, and 10 wt% P-tCNCs, respectively. A neat PU solution without P-tCNCs was also prepared as a control (0 wt% P-tCNCs). The solutions were stirred overnight at room temperature and then heated to 80 °C until the full dissolution of the polymer (2–4 h). Each solution was then transferred into a 20 mL glass-syringe and mounted on a NE-300 Just Infusion syringe pump (New Era Pump Systems Inc., New-York, NY, USA) equipped with a 21G needle. The flow rate of the pump was set to 50 µL/min. A cylindrical collector with a diameter of 7 cm was placed at a distance of 15 cm to the needle tip (Figure 1). A collector rotation speed of 11 m/min was used to produce mats exhibiting a random fiber orientation, while a speed of 330 m/min was used to produce mats with aligned fibers. Using a high-voltage power supply Spellman SL30 (Spellman, Hauppauge, NY, USA), a constant voltage of 8–10 kV was applied between the needle tip and the collector to maintain a stable cone-jet during the entire electrospinning process. The electrospinning process was run for 180 min to obtain sufficiently thick fiber mats (>50 µm). The fiber mats were dried overnight at room temperature in a fume hood to remove any excess DMF.

### 2.4. Density Determination

At least three different samples with an area of 1×1 cm^2^ were cut from each of the 0, 1, 5, and 10 wt% electrospun PU/P-tCNC mats using a razor blade. For each square sample, the mass was determined based on at least three different measurements using a Mettler Toledo ML-T analytical balance (Mettler-Toledo, Greifensee, Switzerland; 0.1 mg precision). The thickness of the mats was determined by micrometer measurements (Mitutoyo MDH high-accuracy sub-micron digital micrometer) taken at at least five different positions across each square sample. The mat density ρ was then calculated using ρ=m/(tA), with *m*, *t*, and *A* the mass, thickness, and area of the sample, respectively.

### 2.5. Scanning Electron Microscopy (SEM)

Nanocomposite fibrous PU/P-tCNC mats were imaged using a Tescan Mira 3 LMH scanning electron microscope at 4 kV (Tescan, Brno, Czech Republic). A 3-nm conductive gold layer was deposited onto the samples to prevent charging, using a 208 HR Cressington sputter coater. The images were analyzed using ImageJ software.

### 2.6. Thermogravimetric Analysis (TGA)

A Mettler-Toledo STAR thermogravimetric analyzer system (Mettler-Toledo, Greifensee, Switzerland) equipped with Al_2_O_3_ crucibles was used for thermogravimetric analysis in a temperature range of 0–600 °C at a heating rate of 10 °C/min under a nitrogen atmosphere.

### 2.7. Atomic Force Microscopy (AFM)

AFM imaging of neat CNCs was performed with a Nano Wizard II (JPK BioAFM, Berlin, Germany) microscope operated in tapping mode using aluminum-coated silicon probes (NanoAndMore) with a nominal force constant of 40 N/m, a nominal resonance frequency of 300 kHz, and a tip radius <10 nm. Samples were prepared by depositing 40 µL of an aqueous CNC solution (0.05 mg/mL) onto freshly cleaved mica substrates and subsequent drying overnight at room temperature. AFM images were processed and analyzed with Gwyddion and ImageJ software.

### 2.8. Tensile Testing

Tensile tests of electrospun PU/P-tCNC nanocomposite mats were carried out using a Zwick Roell Z010 ($F_{\max}$ = 10 kN) material tensile testing machine (Zwick Roell, Ulm, Germany). The instrument was equipped with a 200-N load cell and a 2.5-kN clamp. Using a die-cutting tool and according to ASTM D1708, electrospun mats were cut into 15×38 mm^2^ dogbone-shaped samples with thicknesses ranging from 40 to 70 µm. The testing procedure followed the standard method described by ASTM D1708 but without the use of extensometer. Samples were tested with a strain-rate of 22 mm min^−1^ and a pre-load force of 0.005 N. Fiber mats were tested at least three times, and the corresponding values were averaged.

### 2.9. Wide-Angle X-ray Scattering (WAXS)

WAXS measurements of electrospun mats were performed at the University of Delaware on a Xeuss 2.0 SAXS/WAXS system (Xenocs, Sassenage, France) with the X-ray source operated at a voltage of 50 kV and a current of 0.6 mA. Monochromatic X-rays with a wavelength of 1.542 Å (Cu K radiation) were used to irradiate the samples at a sample-to-detector distance of 72 mm as determined by calibration using a silver behenate standard. Scattering patterns were recorded on a Pilatus detector (comprised of three panels) with a total number of 486×618 pixels using 1-h exposure times.

### 2.10. Rheometry

Rheological experiments were conducted on a TA Instruments AR-G2 rheometer (Waters GmbH, Eschborn, Germany) with a 40 mm parallel plate setup in oscillatory mode. Two milliliters of electrospinning solution (PU/P-tCNC in DMF) were deposited onto the lower plate using a pipette. Frequency sweeps were conducted over a frequency range of ω = 0.5–500 rad s^−1^ at room temperature and at 0.5% strain.

## 3. Results and Discussion

### 3.1. P-tCNCs Characterization

Tunicates were chosen as the cellulose source since tunicate-derived CNCs have higher aspect ratios compared to CNCs extracted from plant-based cellulose. Nicharat et al. showed that the incorporation of 10 wt% tunicate CNCs into polyamide 12 increases the stiffness of the composite to 2.7 GPa, while only 1.7 GPa was achieved when cotton CNCs were used at a similar concentration. This shows the superior mechanical reinforcing properties of tunicate CNCs compared to cotton CNCs [21]. In addition, CNCs extracted using phosphoric acid were shown to be dispersible in polar organic solvents (DMF and DMSO) [15], as well as more thermally stable than CNCs extracted using sulfuric acid hydrolysis. The superior thermal stability of P-tCNCs enable high temperature processing with thermoplastic polyurethanes, at a temperature around 200 °C [28].

P-tCNCs produced via phosphoric acid hydrolysis conserve 95% of their initial mass at 270 °C under air conditions as determined by TGA (Figure 2a). They are significantly more thermally stable than tunicate CNCs extracted using sulfuric acid hydrolysis, which degrade around 180 °C [15]. Statistical analysis of individual, well-dispersed P-tCNCs imaged by AFM gives an average length of 678 ± 408 nm (Figure 2b,c). AFM imaging also shows a population of short P-tCNCs, which may arise from the comparably long hydrolysis reaction time and the relatively high dispersity of the raw tunicate cellulose.

### 3.2. Electrospinning

Solutions with several concentrations of P-tCNCs in PU (0, 1, 5, and 10 wt%) were prepared by first dispersing P-tCNCs in DMF and then adding measured amounts of PU. Before electrospinning, the viscosities of these solutions were determined from rheology measurements. The viscosity of a polymer solution, which is dependent on the polymer molar mass and its concentration, is one of the key parameters controlling the morphology and the diameter of fibers in electrospun mats [46,47]. At low viscosities, the surface tension of the extruded polymer is higher than its viscoelastic force, which destabilizes the polymer jet and results in beaded fibers. By increasing the solution viscosity, the chain entanglement can be increased such that the viscoelastic force overcomes the surface tension of the solution and thus allows the generation of beadless fibers [46,47]. The addition of P-tCNCs to the solution (PU in DMF) is shown to increase the viscosity of the solutions and therefore allows modifying both the diameter and the morphology (beaded vs. beadless fibers) of electrospun fibrous mats, as shown in Table 1 and Table 2 and Figure 3.

The viscosities of PU/P-tCNC solutions in DMF with 0 and 1 wt% P-tCNCs in the PU matrix are relatively low (0.88 and 0.95 Pa·s, respectively). Electrospinning of such low-viscosity solutions resulted in beaded fibers, as shown in Figure 3a,b. Interestingly, solutions containing 5 and 10 wt% P-tCNCs in the PU matrix provide sufficiently high viscosities to form beadless electrospun fibers (Figure 3c,d). To study the effect of fiber alignment, mats were collected on a cylindrical collector rotating at take-up speeds of either 11 or 330 m/min. As depicted in Figure 3, mats in which the nanofibers do not have any preferential alignment were achieved at the lower take-up speed. At 330 m/min the speed is sufficiently high to predominantly align the fibers along the rotational direction. The fiber diameter depends on the viscosity (Table 1 and Table 2) and the collector velocity. The fiber diameters in electrospun mats collected at 11 m/min increased with increasing P-tCNC concentration, with diameters as low as 348 nm for 0 wt% P-tCNC in PU and as high as 749 nm for 10 wt%. For the same P-tCNC concentrations, electrospun mats collected at 330 m/min consisted of thinner fibers, indicating the elongation of fibers when collecting at high rotation speeds.

Since both the morphology and diameter of fibers in electrospun PU/P-tCNC mats depend strongly on the P-tCNC concentration and the take-up speed, the packing of fibers is expected to be different from one concentration to another, thus affecting the density of the fibrous mats. Wan et al. showed that the density of electrospun mats (or their porosity) greatly influences their mechanical properties [48], making comparisons of the mechanical properties of materials with different densities ambiguous. We therefore normalized the measured values of Young’s modulus and tensile strength with the density values determined from thickness (volume) and mass measurements of pre-cut mats (Table 3).

### 3.3. Tensile Testing

To study the influence of the P-tCNC concentration and fiber alignment on the mechanical properties of electrospun fibrous mats, dog-bone-shaped specimen were cut from fiber-mats in the three different ways illustrated in Figure 4. In this way, samples of four different P-tCNC concentrations (0, 1, 5, and 10 wt%) were tested in three distinct fiber arrangements, i.e. fibrous mats with fibers aligned along and perpendicular to the strain direction, or with randomly distributed fibers. The stress–strain behavior determined by tensile testing of these dog-bone-shaped samples is shown in Figure 5. Mats spun from solutions of the same P-tCNC concentration can be directly compared since the densities of the mats are similar (Table 3). The mechanical properties of fibrous mats are dependent on the fiber alignment, for each P-tCNC concentration. Fibrous mats with fibers oriented parallel to the strain direction exhibit Young’s moduli and tenacities that are 2–3 times higher than those with fibers oriented perpendicularly to the strain direction, as shown in Figure 5a. Mats with randomly oriented fibers fall between these two limits.

To compare fibrous mats with different P-tCNC concentrations, the mechanical parameters normalized by the mat density were used to yield specific Young’s moduli and specific tensile strengths, shown in Figure 5b. The specific Young’s modulus and tensile strength increase with increasing P-tCNC concentration, with fibers aligned parallel to the strain direction having the highest values. Neat PU mats are relatively soft, with a Young’s modulus of 12.2 MPa for parallel fiber alignment. As expected, the addition of P-tCNCs increased the Young’s modulus, reaching values as high as 94.6 MPa for a parallel fiber alignment in electrospun fibrous PU/P-tCNC mats with 10 wt% P-tCNCs in PU. These results show that increasing the P-tCNC concentration as well as aligning fibers in the strain direction are beneficial to reinforce PU/P-tCNC electrospun fibrous mats. While a reinforcement of electrospun mats by the mere addition of CNCs has been shown before [39,40,41,42], a further stiffening and strengthening of the mats through fiber alignment has not previously been reported.

### 3.4. Crystalline Orientation

The alignment of fibers improves the mechanical properties of materials [49,50,51]. In addition, the molecular orientation (i.e., the orientation of the crystalline PU domains as well as the P-tCNCs within individual fibers in the electrospun fibrous mats) can further improve the (axial) mechanical properties of the material [52]. In earlier studies on the orientation of CNCs within the fibers of electrospun mats, tCNCs embedded in a poly(ethylene oxide) matrix exhibited uniaxial alignment [42], while low-aspect-ratio CNCs did not show such significant orientation, only local anisotropy over relatively short distances [52]. Here, the degree of orientation of P-tCNCs and PU crystalline domains (hard segments) in electrospun PU/P-tCNC mats was assessed by WAXS measurements (Figure 6).

Figure 6 shows 2D WAXS patterns of PU/P-tCNC nanocomposite mats for a P-tCNC concentration of 10 wt%, consisting of either aligned or non-aligned fibers. Only the [110], [200], and [004] peaks of the P-tCNC crystal structure were observed in these WAXS patterns, while other less intense peaks are hidden by the overlapping, intense PU peak. Each of these three P-tCNC peaks shows an anisotropic intensity distribution for mats with aligned fibers (Figure 6b), indicating a preferred orientation of the P-tCNCs within the electrospun fibrous mat. On the other hand, for the mats with non-aligned fibers, all P-tCNC peaks show a uniform intensity distribution, indicating an isotropic orientation of P-tCNCs within the mat, as shown in Figure 6a.

Scattering profiles were obtained from the 2D WAXS patterns by azimuthal integration, as shown in Figure 7a. The reflections corresponding to CNCs were observed for sufficiently high CNC concentrations. The peaks at *q* = 1.65 Å^−1^ (P-tCNCs) and *q* = 1.46 Å^−1^ (PU) were considered to determine the orientation of P-tCNCs and PU hard segments. The P-tCNC and PU peaks of electrospun fibrous mats with non-aligned fibers did not exhibit any azimuthal dependence, irrespective of the P-tCNC concentration (Figure 7d,e), indicating the entirely isotropic nature of the crystals in these mats. This behavior is in contrast to the electrospun fibrous mats with aligned fibers, where the [200] P-tCNC peak shows an intensity maximum at 180° for P-tCNC concentrations of 5 and 10 wt% in the PU matrix (Figure 7b), corresponding to a preferred orientation of the P-tCNCs within these mats. This observation is particularly interesting as the PU reflection in the very same mats (i.e., those with aligned fibers) was found to be isotropic (Figure 7c), implying an insignificant or possibly entirely isotropic orientation of PU hard segments in electrospun fibrous mats, irrespective of the fiber orientation.

## 4. Conclusions

We demonstrate the electrospinning of fibrous polyurethane (PU) mats containing thermally stable, high-aspect ratio cellulose nanocrystals (P-tCNCs) isolated from tunicates by phosphoric acid hydrolysis. The mechanical properties of electrospun fibrous mats were significantly improved by controlling the fiber alignment within the mats. While the mere addition of P-tCNCs was previously confirmed to stiffen and strengthen fibrous nanocomposite mats, our study shows that the alignment of fibers further improves their mechanical properties and allows the tuning of these properties. More specifically, fibrous PU mats containing 10 wt% P-tCNCs showed a very high stiffness when their fibers were preferably aligned along the strain direction, with a Young’s modulus as high as 94.6 MPa. In contrast, electrospun fibrous mats of identical composition were found to be significantly weaker (Young’s modulus as low as 28.1 MPa) when their fibers were oriented perpendicular to the strain direction. The orientation of the P-tCNCs within fibers was evaluated using WAXS, revealing that P-tCNCs adopt a preferential orientation along the fiber axis, while the PU hard segments appear to remain isotropic. The further reinforcement through fiber alignment was achieved without the need of any further post-processing of the electrospun fibrous mats.

In conclusion, the mechanical properties of nanocomposite fibrous PU mats are strongly dependent on the P-tCNC concentration and on the alignment of the fibers. Superior mechanical properties were demonstrated when fibers are aligned along the strain direction, which coincides with a preferred P-tCNC orientation. This synergistic response indicates that fiber alignment and CNC orientation both contribute to the reinforcement of fibrous PU mats. We envision that this strategy can be extended to other polymer matrices, thereby increasing the utility of fibrous mats in applications such as membrane filtration.

## Figures and Tables

**Figure 1 polymers-12-01021-f001:**
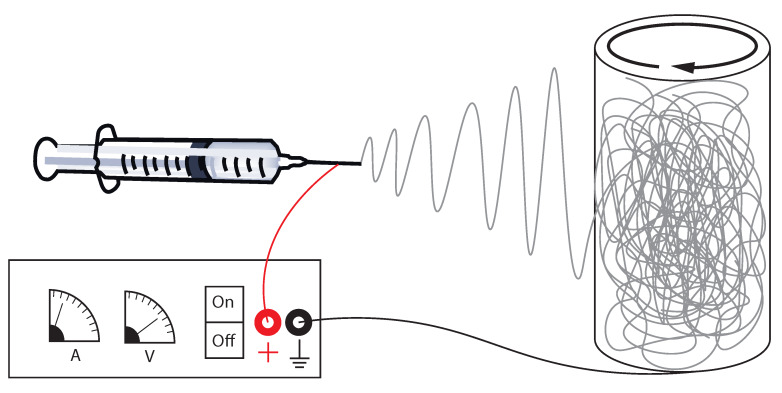
Schematic of the electrospinning set-up used to prepare fibrous PU/CNC mats. A high-voltage power supply is used to apply a voltage of about 10 kV between a syringe that contains a PU/CNC solution in DMF and a cylindrical aluminum collector. Rotation of the collector facilitates production of fibrous mats of homogeneous thickness.

**Figure 2 polymers-12-01021-f002:**
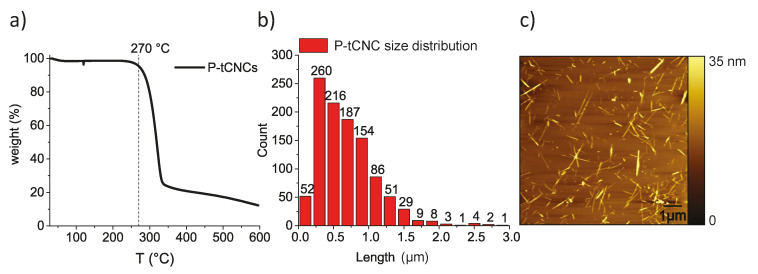
(**a**) The TGA of phosphoric acid-hydrolyzed cellulose nanocrystals from tunicates (P-tCNCs) shows an onset of degradation at 270 °C. (**b**) Size distribution of P-tCNCs as determined from AFM image analysis. (**c**) AFM image of P-tCNCs showing nanocrystals of comparably high aspect ratios.

**Figure 3 polymers-12-01021-f003:**
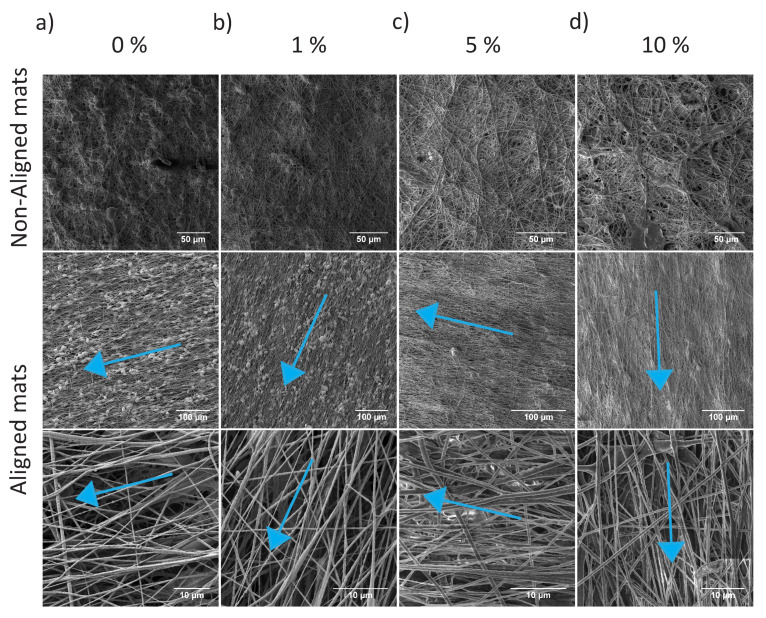
SEM images of electrospun PU/P-tCNC mats collected on a rotating drum at either 11 m/min (top row) or 330 m/min (bottom row), the latter resulting in the alignment of fibers within the mats. The blue arrows indicate the rotational direction of the collector.

**Figure 4 polymers-12-01021-f004:**
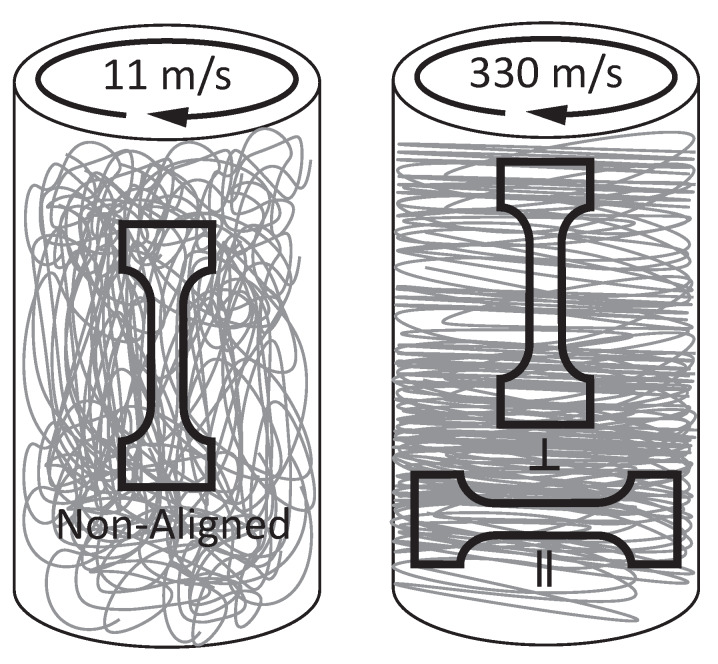
Schematics of dog-bone-shaped electrospun fiber mats illustrating the fiber alignment. Dog bones with randomly deposited fibers (11 m/min), with fibers aligned along (‖, 330 m/min) and perpendicular (⊥, 330 m/min) to the dog-bone axis (strain direction in tensile test experiments). Note that the latter two were cut from identical mats along and perpendicular to the rotational direction of the collector.

**Figure 5 polymers-12-01021-f005:**
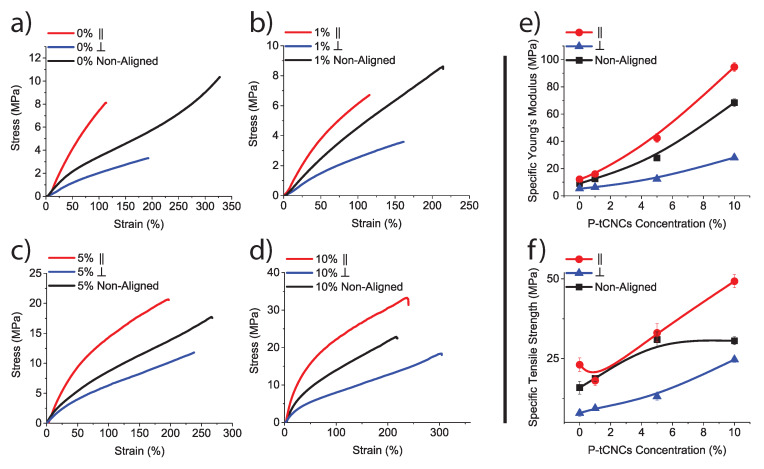
(**a**) Stress–strain curves of electrospun fibrous mats with: (**a**) 0 wt% P-tCNCs; (**b**) 1 wt% P-tCNCs; (**c**) 5 wt% P-tCNCs; and (**d**) 10 wt% P-tCNCs in PU with different fiber orientations, showing the superior mechanical properties of fibrous mats with parallel fiber alignment. (**e**) Specific Young’s modulus and (**f**) specific tensile strength normalized using the densities listed in Table 3, showing an increase in mechanical properties with increasing P-tCNC concentration. Error bars are the same size or smaller than the symbols.

**Figure 6 polymers-12-01021-f006:**
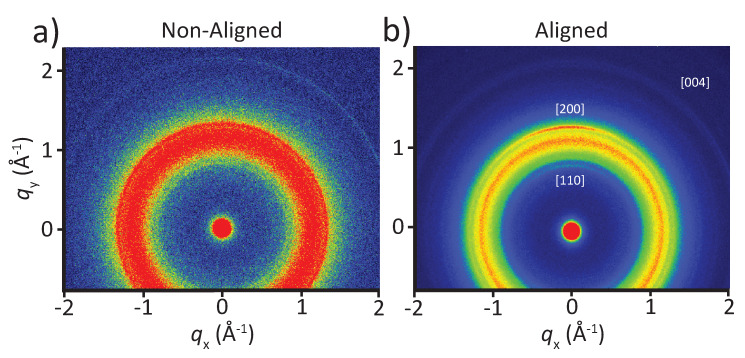
(**a**) 2D WAXS pattern of a 10 wt% P-tCNC nanocomposite electrospun fibrous PU/P-tCNC mat with non-aligned fibers, showing isotropic scattering as indicated by the presence of a continuous ring of uniform intensity. (**b**) 2D WAXS pattern of a 10 wt% P-tCNC nanocomposite electrospun fibrous PU/P-tCNC mat with aligned fibers showing an anisotropic intensity distribution for the [200] as well as (less visible) [110] and [004] peaks of P-tCNCs.

**Figure 7 polymers-12-01021-f007:**
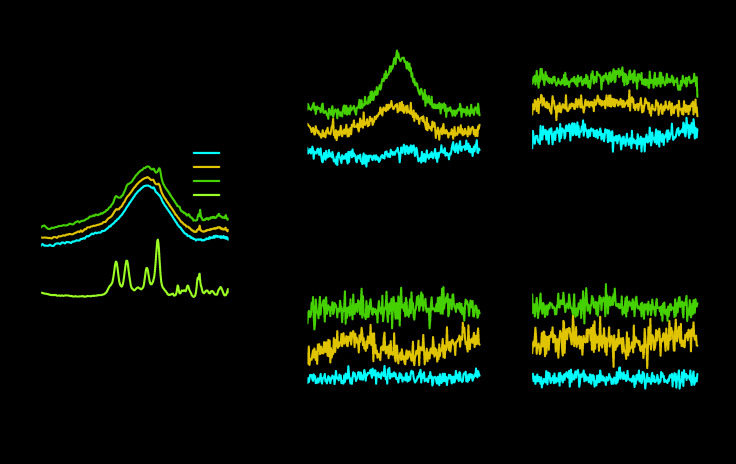
(**a**) Scattering profile of aligned nanocomposite mats, showing an increase in P-tCNC peaks with increasing concentration. (**b**,**d**) Azimuthal profiles of the P-tCNC [200] reflection (*q* = 1.65 Å^−1^) for “aligned” and “non-aligned” mats. A preferential P-tCNC orientation in electrospun fibrous mats with aligned fibers is indicated by the presence of a maximum in the angular-dependent intensity, whereas an isotropic orientation is found by its absence. (**c**,**e**) Azimuthal profiles of the PU WAXS peak (*q* = 1.46 Å^−1^) for “aligned” and “non-aligned” electrospun fibrous mats, revealing the isotropic nature of PU in both cases.

**Table 1 polymers-12-01021-t001:** The viscosity of PU/P-tCNC solutions in DMF increases with increasing P-tCNC concentration.

P-tCNC Conc.	0 wt%	1 wt%	5 wt%	10 wt%
Viscosity (Pa·s)	0.88	0.95	1.57	1.96

**Table 2 polymers-12-01021-t002:** Fiber diameter in electrospun PU/P-tCNC mats based on SEM image analysis. The fiber diameter increases with increasing P-tCNC concentration. Fibers collected at 330 m/min exhibited smaller diameters than those collected at 11 m/min.

Fiber Diameter (nm)					
	P-tCNC Conc.	0 wt%	1 wt%	5 wt%	10 wt%
Drum Velocity					
11 m/min		348±88	460±127	595±153	749±310
330 m/min		274±80	330±96	496±135	578±204

**Table 3 polymers-12-01021-t003:** Densities of electrospun PU/P-tCNC nanocomposite fibrous mats with different fiber alignments (no alignment, ‖, and ⊥). The mat density increases with increasing P-tCNC concentration (weight fraction of P-tCNCs in PU).

Fiber Diameter (nm)					
	P-tCNC Conc.	0 wt%	1 wt%	5 wt%	10 wt%
Drum Velocity					
no alignment		0.66±0.03	0.40±0.02	0.50±0.03	0.72±0.01
‖ alignment		0.39±0.01	0.36±0.03	0.59±0.03	0.69±0.02
⊥ alignment		0.38±0.05	0.40±0.01	0.61±0.02	0.73±0.02

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
