# Peer review of "Electrospinning of Cellulose Nanocrystal-Reinforced Polyurethane Fibrous Mats"

_polymers, 2020, doi:10.3390/polym12051021_

Round 1

Reviewer 1 Report

Major comments:

  1. The author should clarify the Originality / Novelty of this study.
  2. A broader significance of the study to the general audience is missing
  3. The rationale for doing rheological experiments are not clear.
  4. The standard used for mechanical experiments is missing.

Specific comments:

Originality / Novelty

Earlier studies by several authors found Cellulose Nanocrystal reinforced Polyurethane. For example, Pei et al. (Macromolecules, 2011) found strong Nanocomposite Reinforcement Effects in Polyurethane Elastomer with Low Volume Fraction of Cellulose Nanocrystals. This study also cited Santamaria-Echart et al. (Carbohydrate polymers, 2016) works where it was found that Cellulose nanocrystals (CNC) reinforced polyurethane nanocomposites. The effect of fiber alignment and load direction on the mechanical strength of polymeric material is well established. Therefore,

Significance of Content

This study demonstrated the electrospinning of nanocomposite fibrous mats comprised of a PU reinforced with t-CNCs improved mechanical properties of electrospun fibrous mats upon CNC addition. This study also confirmed that the alignment of fiber influenced stiffness and strength of fibrous PU/CNC nanocomposite mats. This study is significant as it widens the range of the composite application in wound dressing or filtration technology.

Quality of Presentation

This study presented adequately the influence of CNC addition on the fiber morphology and diameter. A drum-type fiber collection method was used to produce aligned or isotropically oriented fibers by varying the speed of the drum.  The rationale for doing rheological experiments is not clear. It appears that rheological experiments were used to determine solution viscosity for producing fiber mat. A figure or table showing how the viscosity influences mechanical strength can increase the quality of the presentation.

Scientific Soundness

Mechanical properties were evaluated by tensile testing. But the author didn’t use or mention the ASTM standard used for mechanical testing.

Reviewer 2 Report

In this manuscript, the authors studied the mechanical properties of CNC reinforced PU nanocomposite fibrous mats and found that the mechianical properties are strongly dependent on the P-tCNC concentration and the alignment of the fibers. The manuscript is well written. However, the novelty and motivation of this work is not clearly explained. It is only recommended for publication after the following comments are addressed.

(1) P1L21-22, the authors stated that “However, due to their non-woven nature, electrospun mats are mechanically weak, restricting their direct use in certain applications.” Please list some application examples here.

(2) CNC has been widely used to reinforce electrospun polymer fibers and PU/CNC electrospun fibers have already been studied (e.g., Arantzazu Santamaria-E chart et al. Carbohydrate Polymers, 2017, 166, 146-155; Li Zhu et al. ACS Appl. Mater. Interfaces 2019, 11, 13, 12968-12977). Furthermore, it is a kind of common sense that nanofiller concentration and fibers alignment play an important role to the mechanical properties. What's the novelty and motivation of this work? Please explain in the indrodution.

(3) How many samples were tested for tensile testing? From the data shown in Figure 5, only one sample was tested for each group. For mechanical property testing, triplicates are necessary. Average and SD should be provided.

Round 2

Reviewer 1 Report

Although the response to my comments is satisfactory, this study locks scientific significance as the technology is limited to specific material. The authors must add a paragraph in the discussion section on how this research can be broadened and translated to future research that can benefit the electrospun research community and the technology can be used in the polymer industry with a specific application. The abstract must include a sentence showing the novelty of this study.

The readability of the manuscript is poor. A deep and accurate English grammar revision is required.

Reviewer 2 Report

The authors addressed all of my concerns. The manuscript is recommender for publication now.

Author Response

We thank the reviewer for taking the time to review our manuscript and are glad about to positive feedback.